# The Protein Interactome of Glycolysis in *Escherichia coli*

**DOI:** 10.3390/proteomes9020016

**Published:** 2021-04-06

**Authors:** Shomeek Chowdhury, Stephen Hepper, Mudassir K. Lodi, Milton H. Saier, Peter Uetz

**Affiliations:** 1Integrative Life Sciences, Virginia Commonwealth University, 1000 West Cary Street, Richmond, VA 23284, USA; schowdhury2@vcu.edu or; 2Center for the Study of Biological Complexity, Virginia Commonwealth University, Richmond, VA 23284, USA; shepper@mymail.vcu.edu (S.H.); lodimk@mymail.vcu.edu (M.K.L.); 3Department of Molecular Biology, Division of Biological Sciences, University of California at San Diego, La Jolla, CA 92093, USA; msaier@ucsd.edu

**Keywords:** protein, glycolysis, interaction, *Escherichia coli*, protein-protein interaction/PPI

## Abstract

Glycolysis is regulated by numerous mechanisms including allosteric regulation, post-translational modification or protein-protein interactions (PPI). While glycolytic enzymes have been found to interact with hundreds of proteins, the impact of only some of these PPIs on glycolysis is well understood. Here we investigate which of these interactions may affect glycolysis in *E. coli* and possibly across numerous other bacteria, based on the stoichiometry of interacting protein pairs (from proteomic studies) and their conservation across bacteria. We present a list of 339 protein-protein interactions involving glycolytic enzymes but predict that ~70% of glycolytic interactors are not present in adequate amounts to have a significant impact on glycolysis. Finally, we identify a conserved but uncharacterized subset of interactions that are likely to affect glycolysis and deserve further study.

## 1. Introduction

Proteomics has produced invaluable data about the composition, abundance, and function of all proteins in cells and organisms [1]. Similarly, interactomics has produced insights into the physical interactions among those proteins in cells [2,3]. However, although these studies can provide functional insights, most proteins and especially enzymes need individually designed experiments for a full understanding, even though some attempts have been made to study enzymes on a large scale [4,5].

Glycolysis is the primary central carbon metabolic pathway responsible for turning glucose into pyruvate (especially under anaerobic conditions), besides producing ATP and other metabolites (Figure 1). Glycolysis is catalyzed by 14 enzymes (Table 1) which are regulated by allosteric interactions [6], post-translational modifications [7], changes in gene expression, and protein localization [6]. However, only a few PPIs that regulate glycolysis are known (Table 2 and below). In fact, we found several new interactions affecting glycolysis only recently. For example, the small phosphocarrier protein (HPr) of the bacterial phosphotransferase system globally regulates energy metabolism by directly interacting with pyruvate kinase (PykF, but not PykA) and phosphofructokinase (PfkB, but not PfkA), but also glucosamine-6-phosphate deaminase (NagB), and adenylate kinase (Adk). This last interaction allows HPr to regulate the cellular energy charge coordinately with glycolysis [8,9]. In addition to these examples, there are numerous other interactions that affect glycolysis indirectly, e.g., enzymes that feed directly (or indirectly) into glycolysis [8,10,11].

It may be surprising that hundreds of proteins are known to interact physically with practically all glycolytic enzymes, yet their impacts on glycolysis are virtually unexplored. In fact, despite numerous proteomic and interactomic studies in *E. coli* and other bacteria, there are few comprehensive analyses of the interface of the protein interactome and the metabolome in this or other species [12,13].

Here we explore this interface in *E. coli*, specifically the interactome of glycolytic enzymes with other proteins not yet known to play a role in this pathway. Besides summarizing our knowledge on glycolytic PPIs, we investigate the role of protein abundance and thus stoichiometry as well as phylogenetic conservation to identify those interactions which are most likely physiologically relevant and thus deserve experimental validation.

This study will also pave the way for future studies of enzyme regulation involving post translational modifications (PTMs) and gene expression, both of which are critical in metabolic regulation. Allosteric regulation is another mechanism that has been tackled experimentally, e.g., in Hpr, GlnB, and NanE, all of which allosterically regulate multiple enzymes feeding into glycolysis [8,9]. If we understood glycolysis in all its details, this knowledge may be useful in many applications, ranging from metabolic engineering to targeting enzymes by antimicrobials [14,15,16,17].

**Table 2 proteomes-09-00016-t002:** Previously found protein-protein interactions of glycolytic enzymes.

Glycolytic Enzyme	Interactor	Reference
Eno	CsrA	[18]
GapA	GlnA	[19]
GapA	NagB	[20]
GpmA	GpmI	[21,22]
GpmI	NagB	[8]
PfkB	PtsH (Hpr)	[9]
PfkB	Zwf	[9]
Pgk	Rne	[23]
PykF	PtsH (hpr)	[9]

## 2. Materials and Methods

### 2.1. Protein and Interaction Data

Protein data for *E. coli K12* was obtained from its Uniprot reference proteome (ID: UP000000625), consisting of 4391 proteins, including 14 glycolytic enzymes. These enzymes catalyze the nine steps of glycolysis (Table 1). While we initially used various quantitative proteome datasets from PaxDB [24], we eventually decided to do our analysis with data from Schmidt et al. [1] because it is one of the most comprehensive proteome datasets for *E. coli* and it is one of the few studies that use various carbon sources that feed (or do not feed) into glycolysis, so it was the most appropriate data for our study.

#### 2.1.1. PPIs from Intact

The IntAct database was searched for binary interactions using the 14 glycolytic enzymes (Table 1). This resulted in 402 binary interactions from which all spoke-expanded interactions were removed, which resulted in 97 binary interactions (“spoke expansion” assumes that proteins which have been co-purified with a particular bait protein interact with that bait). Also, the 97 interactions included chemical molecules/metabolites as their interactors decreasing the number of actual protein interactors to <97. After exploring the literature, we found five published studies that contained large-scale interaction data, namely those by Butland, Hu, Rajagopala, Lasserre and Häuser et al. (Table 3).

#### 2.1.2. PPIs from String

The STRING database was searched for binary interactions using the 14 glycolytic enzymes from *E. coli* (Table 1). STRING differentiates its interaction data into sources such as text mining, co-expression, databases, and experiments etc. and assigns scores to those interactions in the range of 0 to 1. However, the interaction scores were comparatively low for the interaction source “experiment” (below 0.5) whereas scores above 0.9 were assigned for other sources such as co-expression etc. Examining the literature, we found **nine** published studies including Rajagopala and Hu totaling to 11 large-scale unique [14 = IntAct (5) + String (9)] studies that provided the PPI data (Table 3).

We obtained all PPIs involving 14 glycolytic enzymes from the datasets associated with these 11 articles resulting in a final list of 352 interactions. After removing duplicates, we obtained a list of 339 PPIs from these 11 published studies (Appendix A, Table 3).

### 2.2. Published Evidence for Protein-Protein Interactions Regulating Metabolism

In order to find more detailed studies on PPIs regulating glycolysis, we downloaded all the papers from PUBMED mentioning the keywords *Escherichia coli* and glycolysis. Second, we filtered this set for papers with two glycolytic proteins/gene names in the same abstract. For example, pykF (glycolytic enzyme) and ptsH (interactor; also known as HPr) were found in the abstract of Rodionova et al. [9]. We considered those interactions as *evidence-based interactions*. In this manner, we obtained 64 evidence-based “interactions” (58 w/o duplicate interactions) out of the total of 352 (339 w/o duplicates) PPIs Appendix A). These 58 “interactions” were manually examined further and a total of nine physical PPIs identified (Table 2). The remaining ones were often indirect (such as regulatory interactions without further mechanism) or otherwise lacking details of a physical interactions. We have performed our analyses using these two sets of interactions, i.e., all (352/339) and evidence-based (64/58) PPIs.

### 2.3. Duplicate Interactions in Multiple Sources

Twelve PPIs were mentioned in two large-scale studies such as those by Hu et al. [3] and Rajagopala et al. [2]. Large-scale studies have a certain fraction of false positives and thus require further validation but when PPIs are found in more than one such study they are much more reliable [27]. One of these interactions i.e., enolase binding rne (ribonuclease E) was mentioned in three studies (Appendix A, Table 4).

### 2.4. Properties of Glycolytic Interactors

In order to identify enzymes among glycolytic interactors, we used the EC number in the Uniprot reference proteome [34]. Similarly, we used the Uniprot GO (Gene Ontology) terms to map proteins to biological processes, including “metabolic process” to find metabolic proteins or proteins not known to be involved in any metabolic *pathway*. This information was verified using annotations from KEGG [35] to identify proteins involved in carbohydrate metabolism or other processes, such as environmental information processing. Lastly, we identified essential genes/proteins using the Database of Essential Genes (DEG version 15.2) [36].

### 2.5. Conservation and Abundance of Glycolytic Interactors

In order to investigate the phylogenetic conservation of proteins, we used the EggNOG database to find the number of genomes in which a protein is present. This number gives a good estimate of how conserved a protein is across the phylogenetic tree (Section 3.2). The abundances of proteins were derived from Schmidt et al. [1] who determined protein abundances as copies per cell under various growth conditions, e.g., in glucose vs. pyruvate (as carbon source). We used glucose as a reference condition (see Section 3.3).

**Figure 1 proteomes-09-00016-f001:**
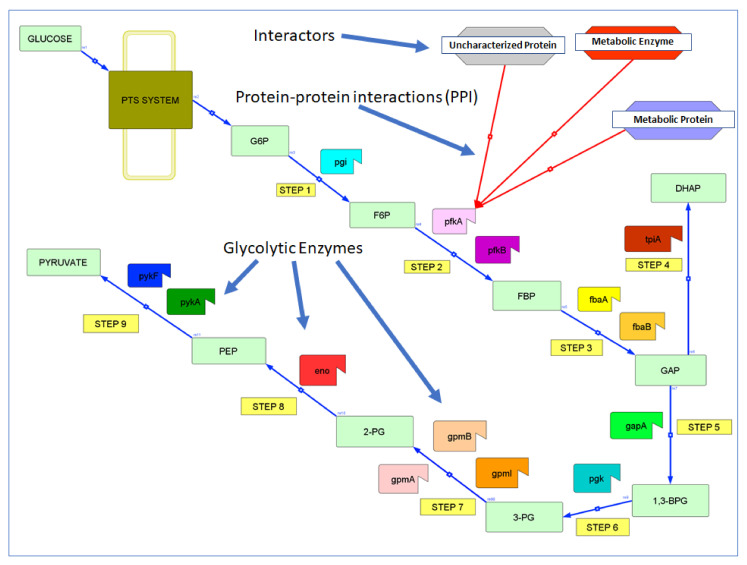
Glycolysis in *E. coli K12*. The 14 glycolytic enzymes interact (red arrows) with interactors (E.g., Metabolic Enzyme). For abbreviations of enzymes and substrates see Table 1. The pathway was drawn with Cell Designer [37].

## 3. Results

The 14 glycolytic enzymes in *E. coli* interact with at least 237 unique proteins in at least 339 interactions. These proteins also involve 14 proteins of unknown function (Appendix A). Both data and results are summarized in Table 5.

### 3.1. Glycolytic Enzymes Interact Very Specifically with Non-Glycolytic Proteins

The 14 enzymes have very distinct interaction patterns with some proteins having almost none and some proteins having dozens of interactions (Figure 2A). For instance, Eno has 93 interaction partners including five uncharacterized proteins (UPs). By contrast, GpmB (predicted phosphoglycerate mutase) has only one interaction. All other enzymes had interaction numbers in between these extremes (Table 1). Overall, 16 uncharacterized proteins interacted with glycolytic enzymes and one of these interactions, PykA:YggR (a putative pilus biogenesis ATPase), has been found in two studies (Table 4). One UP (YdjL; a putative zinc-type alcohol dehydrogenase) was found to interact with two glycolytic enzymes, namely with enolase and GapA. Hence, the number of unique uncharacterized proteins in our dataset is 14.

Six proteins have uncharacterized interaction partners, namely Pgk (1 interactor), GpmA (1), GpmI (2), PykA (2), GapA (4), and Eno (5). These numbers are roughly proportional to their total numbers of interactions (Eno: 90 unique interactors, GapA: 56, GpmA: 13 GpmI: 4).

Among the interactions reported previously, again Eno and GapA had the most interaction partners with 13 and 10 interactors, respectively (Figure 2B). The other enzymes have from 0 to seven reported interactions in the literature.

**Figure 2 proteomes-09-00016-f002:**
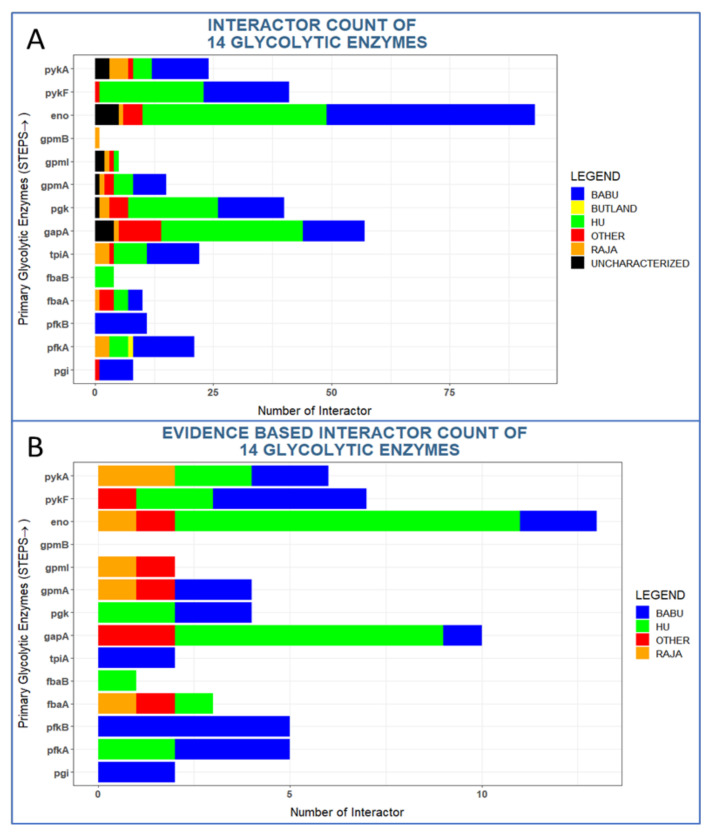
Interacting partners of glycolytic enzymes in *Escherichia coli* and their main sources. The 6 different color codes in the plot correspond to the 4 large-scale PPI studies and 7 small scale data sets (“other”, Table 3). Uncharacterized proteins are shown in black. Evidence-based interactions have support from text-mining (see methods).

#### 3.1.1. Glycolytic Enzymes Interact with Both Enzymatic and Non-Enzymatic Proteins

We wondered if glycolytic proteins interacted with other enzymes or rather with non-enzymatic proteins, such as catalytically inactive regulators. As a proxy for enzymatic activity, we used the EC number provided by Uniprot. Overall, glycolytic enzymes interact with both enzymes as well as non-enzyme proteins at similar ratios (46.8% enzymes vs. 47.2% non-enzymes with 5.9% uncharacterized, Figure 3A).

For instance, enolase (Eno), the most promiscuous enzyme in glycolysis interacts with 49 enzymatic protein partners and 39 non-enzymatic protein partners as well as 5 uncharacterized proteins. All others had between 0 (GpmB) and 27 (GapA) enzyme interaction partners. Notably, PykF was the only glycolytic enzyme which had more non-enzyme partners than enzymes. The preference for enzymatic partners is even more pronounced among the literature-curated interactions (Figure 3B): 73.9% of glycolytic interactors are enzymes vs. 26.1% being non-enzymes. However, three enzymes (FbaB, TpiA, and PykF) had more non-enzymatic partners than the others.

#### 3.1.2. Glycolytic Enzymes Interact with Both Metabolic and Non-Metabolic Proteins

Given that many glycolytic enzymes interact with non-enzymes, we wondered how many interactors were involved in other metabolic pathways or in non-metabolic pathways. The metabolic functionality of a protein was determined using Gene Ontology (GO) Biological Process annotations (from Uniprot). If the interactor had at least 1 GO term mentioning “metabolism”, we considered it to be a metabolic protein. Note that the GO classification has a broad definition of “metabolism” so that processes like DNA repair are also considered “metabolic” (because they are involved in “DNA metabolism”).

Overall, 50.2% of glycolytic interactors were involved in metabolism, with carbohydrate GO metabolic terms like glucose metabolic process [GO:0006006]/carbohydrate metabolic process [GO:0005975]- being the most common with 27 interactors. Remaining 210 interactors showed a wide range of GO metabolic terms including ribosomal large subunit assembly (translation), ATP biosynthetic process, L-proline biosynthetic process (amino acid metabolism) etc. There were 25 proteins which did not show any GO metabolic terms (Figure 3C,D).

For instance, Eno had 35 metabolic protein interactors, 43 non-metabolic protein interactors, 7 interactors which did not show any GO terms and 5 uncharacterized proteins. The most-common metabolic GO terms among enolase’s interactors were cellular response to DNA damage stimulus [GO:0006974] and mRNA catabolic process [GO:0006402]. Interestingly, enolase had more non-metabolic interactors, and the most over-represented (high-level) GO terms dealt with regulation of cell shape [GO:0008360], peptidoglycan biosynthetic process [GO:0009252] and cell wall organization [GO:0071555] (43 interactors).

The evidence-based analysis reflected the pattern we saw in the enzyme vs. non-enzyme distribution, with most enzymes having metabolic interactors. FbaB was the only glycolytic enzyme that had only non-metabolic interactors, namely DnaJ (chaperone/heat shock protein J).

Overall, more metabolic than non-metabolic interactors were found in the glycolytic interactome, as expected, but the substantial number of non-metabolic interactors indicates an extensive crosstalk between glycolysis and other processes in bacterial cells.

We used the KEGG database to determine the pathway characteristics of the interactors (Figure 3E,F). Our analyses showed 22 proteins were part of a carbohydrate metabolic pathway, 94 interactors were part of a non-carbohydrate metabolic pathway and 121 interactors were not involved in any metabolic pathway including 14 uncharacterized proteins. For instance, Eno had nine interactors involved in carbohydrate metabolic pathways, 40 interactors involved in non-carbohydrate metabolic pathways, 42 had no metabolic pathways, and five were uncharacterized proteins.

#### 3.1.3. Glycolytic Enzymes Interact Primarily with Non-Essential Genes

While glycolysis is an important central pathway, it is not absolutely essential, as long as cells can obtain energy from sources other than glucose, such as amino acids. However, if glucose is the only carbon source, glycolysis becomes an absolutely essential pathway, hence it is conditionally essential. We wondered if glycolytic enzymes interact with other essential proteins or whether these interactors are dispensable.

We evaluated if a protein is encoded by an essential gene using the Database of Essential Genes (DEG). Then, we counted the number of essential genes and non-essential genes among the interactors of glycolytic enzymes. Uncharacterized proteins are not essential, as expected. These proteins did not have an EC number (except 1 protein: YegV), and did not show metabolic terms from GO or KEGG (metabolic pathways).

In the raw data analysis, glycolytic enzymes generally interacted with more non-essential than essential proteins, and Pgi, GpmI and GpmB had *only* non-essential proteins as interactors (Figure 3G). As the most promiscuous protein, enolase shows a fairly representative pattern with 29 essential gene interactors, 59 non-essential gene interactors and 5 uncharacterized proteins. In the evidence-based analyses, the pattern was similar, although there were no uncharacterized interactors (Figure 3H).

#### 3.1.4. Glycolytic Enzymes Interact Primarily with Metabolic Proteins

Once we had identified interactions with enzymes and metabolic proteins, we wanted to differentiate these proteins further, especially with regard to metabolic pathways. Hence, we defined 4 types of proteins based on the aforementioned results (Table 6): Metabolic Enzyme (ME): An interactor that is an enzyme (based on EC number) AND a metabolic protein (from a metabolic GO term) or part of a metabolic pathway (from metabolic pathway KEGG term). Metabolic Protein (MP): An interactor that is NOT an enzyme (EC number) AND is a metabolic protein (GO) or part of a metabolic pathway (KEGG). Non-Metabolic Enzyme (NME): An enzyme (EC) AND NOT a metabolic protein (GO) AND NOT part of a metabolic pathway (KEGG). Non-Metabolic Protein (NMP): A non-metabolic protein AND NOT an enzyme (EC) AND NOT a metabolic protein (GO) AND NOT part of a metabolic pathway (KEGG).

Using this classification, approximately 70% of the all the interactors of 14 glycolytic enzymes are parts of metabolism (Figure 4).

### 3.2. Many Interactors of Glycolytic Enzymes Are Highly Conserved

Glycolysis is a fundamental process found in all domains of life; hence glycolytic enzymes are highly conserved. We hypothesize that important interactors should be equally well conserved even though they could be conserved for other reasons. We found that glycolytic interactors are present in a large number of bacterial genomes ranging from 100 to 4000 out of 10,000 bacterial genomes [38] with an average of 2635 genomes (Figure 5). For instance, enolase has 35 interactors found in more than 3000 genomes but also 6 interactors found in fewer than 1000 genomes. Overall, glycolytic interactors seem to be better conserved than the average protein in the *E. coli* proteome, but much less conserved than any of the glycolytic enzymes which are almost universally conserved across bacteria.

### 3.3. Most Glycolytic Enzymes Are More Abundant than Their Interactors

Even if a protein binds to an enzyme with high affinity, the interactor can affect enzyme activity only when it is present at nearly stoichiometric (or higher) levels. Although there is little experimental data supporting this assumption for in vivo levels of enzymes, we investigated the expression levels of glycolytic interactors under different conditions as described by Schmidt et al. [1].

We normalized the abundance values of all interactors to the level of the glycolytic enzyme (set to 1). In general, the abundance of most interactors is lower than that of their corresponding glycolytic enzymes (Figure 6). That is, only interactors that are present at a similar or higher stoichiometric level than their enzymes are able to affect their activities in a notable way. Unfortunately, binding affinities are only known for very few such interactions; thus, it remains impossible to quantitatively assess the actual impact on those enzymes. In fact, one of the goals of this study was to identify PPIs that are top targets for such quantitative measurements (see below).

## 4. Discussion

Metabolic pathways are related to many diseases, and many therapeutic applications cannot be properly developed if an adequate understanding of their regulation is not available. However, research has focused mainly on allosteric regulation, gene regulation and post-translational modifications. Protein-protein interactions (PPIs) can also be a very important route in terms of affecting metabolic enzymes [8,9,10]. Here we report our studies revealing the potential impacts of PPIs on glycolysis.

In order to get a more comprehensive picture of glycolytic PPIs, we visualized the glycolysis PPI network (Figure 7). The network shows that glycolytic enzymes physically interact with interactors of different metabolic pathways and processes such as carbohydrate metabolism, lipid metabolism, nucleotide metabolism and amino acid metabolism, but also with enzymes of DNA replication and repair, mRNA translation and glycan biosynthesis. Note that replication and repair, translation and transcription as well as protein folding, sorting and degradation are considered as (metabolic) pathways by KEGG, and many of these processes could be regulating glycolysis or vice versa. In fact, such “crosstalk” between pathways has long been known [39] even though many details remain unclear.

The flux of metabolic reactions is determined by the concentrations of enzymes and substrates, but also by their activities and their rate-limiting steps. Hence, we explored the abundance of the glycolytic proteome with respect to its interactome (Figure 7). We found that the abundances of glycolytic enzymes may be higher or lower than the abundances of their interactors (Figure 6). Even if an interactor binds with high affinity to an enzyme, that interaction will only have a noticeable impact on the enzyme if the interactor is present at a sufficient concentration. Hence, we estimate that ~70% of glycolytic interactors are not present in adequate amounts to have a significant impact on glycolysis. Interestingly, the concentration would be particularly critical for the rate-limiting enzymes, namely Pfk and Pyk [41], both of which are present as two isoforms (Figure 6). For a precise estimate, however, we would need measurements of binding affinity which are not available for the vast majority of interactions.

Despite the almost universal conservation of glycolysis throughout all domains of life, many of the interactors considered in our analyses are much less well conserved (Figure 5). While clade-specific proteins may fulfil important roles, adapted to specific circumstances, we can safely assume that broadly conserved proteins may play other important roles. We found that 48.5% of glycolytic interactors are conserved in more than 2871 (median) bacterial genomes (or 42.6% in more than 3000 genomes), and we thus focused our attention on these conserved proteins (see below).

In order to prioritize interacting proteins for further study, we have identified a starting set of 9 proteins that are potential regulators of glycolysis in *E. coli* and other bacteria (Table 7). Many others will deserve further investigation but we refrained from singling them out at this point (Appendix A).

Genetic experiments can help us to validate these candidate interactions, especially if we can mutate the interaction sites without affecting the enzymatic (or other) activities of these proteins. For example, knocking out the binding site on rne (Ribonuclease E) may increase or decrease the metabolic rate of *E. coli* without affecting RNA metabolism. This approach will further establish the fact that protein-protein interactions regulate metabolic pathways in critical ways.

## Figures and Tables

**Figure 3 proteomes-09-00016-f003:**
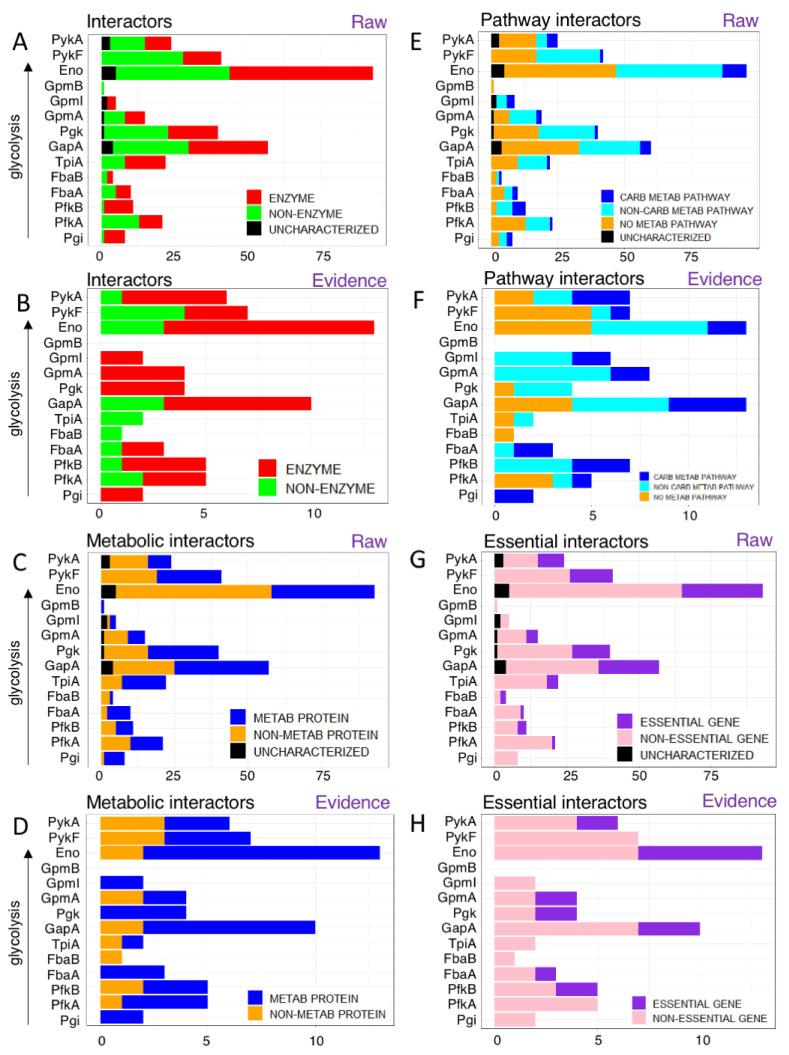
What do glycolytic enzymes interact with? (**A**) Most interacting partners of glycolytic enzymes are enzymes, but 47.3% are not. Note that the raw datasets (**A**) had also 14 uncharacterized interactors (black) while the evidence-based set did not (**B**). (**C**) The number of metabolic and non-metabolic protein partners. The glycolytic interactome is dominated by non-metabolic interactors (44.7%) although primarily enzyme interactions have been characterized further in the literature (**D**). Note that no uncharacterized proteins were involved in interactions of evidence-based PPIs while 14 such proteins involved in 15 PPIs are in the raw data (**C**). (**E**) Glycolytic enzymes interact with proteins of many pathways. Carbohydrate metabolism (blue) is the most common but overall non-carbohydrate metabolic pathways (cyan) are more common, with a substantial number of interactions from non-metabolic (orange) or uncharacterized proteins (black). (**F**) Evidence: even in the literature-curated dataset, carbohydrate pathways are under-represented. (**G**,**H**) Glycolytic enzymes interact primarily with non-essential proteins. Only the raw data (**G**) had some uncharacterized proteins. (**H**) Evidence-based glycolytic PPIs.

**Figure 4 proteomes-09-00016-f004:**
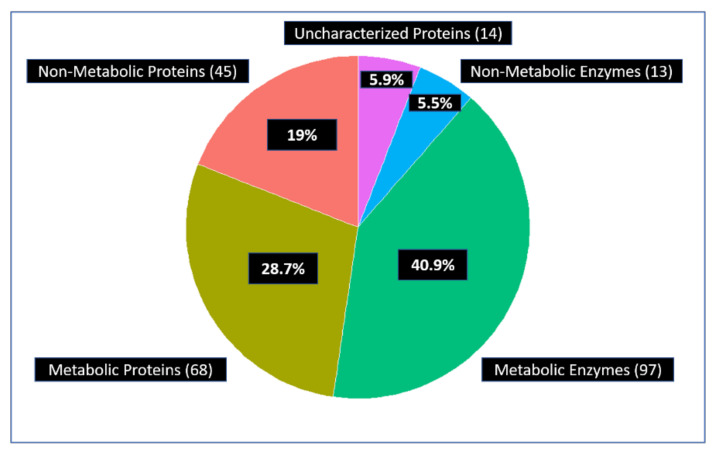
About 70% of the 237 unique interactors are metabolic enzymes or other metabolic proteins. These include 97 metabolic enzymes and 68 metabolic proteins but also regulators that control metabolism. The others are 45 non-metabolic proteins, 13 non-metabolic enzymes, and 14 uncharacterized proteins.

**Figure 5 proteomes-09-00016-f005:**
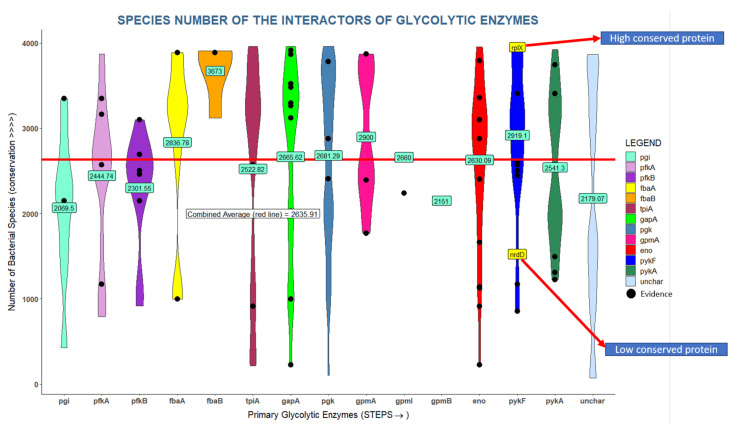
Many glycolytic interactions are highly conserved across bacterial genomes. The glycolytic enzymes are arranged on the X axis with their interactors violin-plotted in columns. The Y axis shows number of genomes in which an interactor is found. Interactors are color coded by primary glycolytic enzyme with which they are interacting (pgi interactors = cyan). Interactors in black are supported by literature reports. The average conservation value across all interactors is ~2636 genomes (red line). Numbers in green boxes are the averages across interactors of each glycolytic enzyme.

**Figure 6 proteomes-09-00016-f006:**
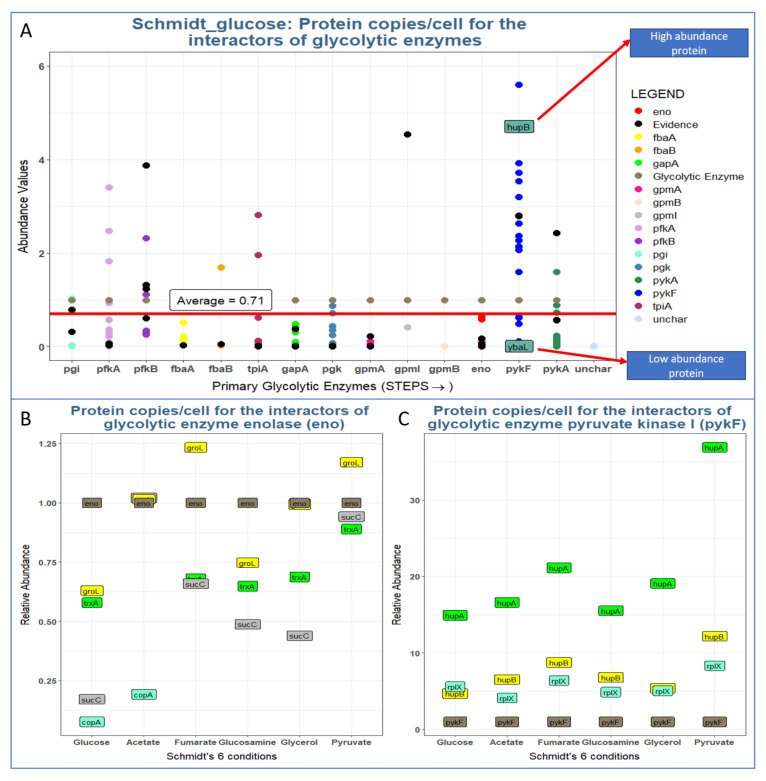
Does the expression level of glycolytic interactors have an impact on glycolysis? Most interactors are expressed at sub-stoichiometric levels and thus will likely not have a strong impact on its target glycolytic enzyme. (**A**) Glycolytic enzymes (normalized to relative level = 1) and the stoichiometry of their interactors (average protein count = 0.7 copies for each copy of its corresponding glycolytic enzyme). The order of the enzyme appearance on the *x* axis is the same as the order of the enzymes appearing in glycolysis. (**B**,**C**) Condition-specific stoichiometries of enolase (**B**) and pyruvate kinase (**C**) and their interactors under six different conditions (single carbon sources = glucose, acetate, etc.). Glycolytic enzyme abundance values are normalized to 1. Protein expression data are from Schmidt et al. 2016 [1].

**Figure 7 proteomes-09-00016-f007:**
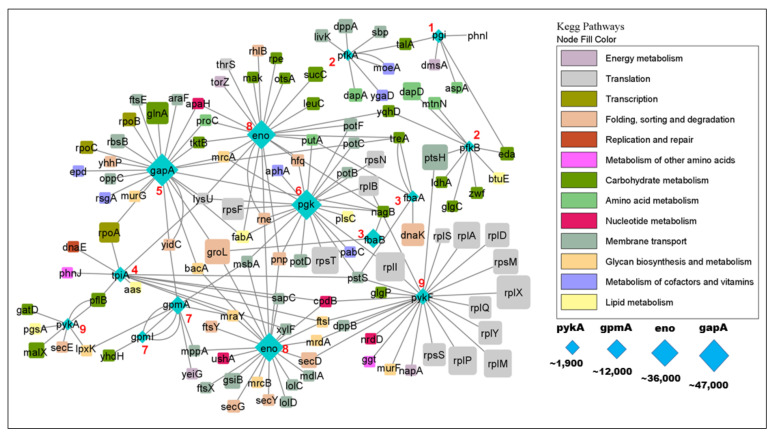
The glycolytic interactome. Glycolytic enzymes are shown in blue, with the step in glycolysis shown as a red number. Functional categories of interactors are color coded as shown on the right. Protein abundance (copies per cell) is proportional to the symbol size except for ribosomal proteins whose sizes are capped (otherwise they would vastly outsize all other proteins). Note the tight integration of glycolysis with other metabolic pathways such as carbohydrate, lipid, and nucleotide metabolism. Compare to Figure 2 and Figure 6. Network created in Cytoscape 3.8.2 [40].

**Table 1 proteomes-09-00016-t001:** The 14 metabolic enzymes controlling the 9 metabolic reactions of glycolysis in *Escherichia coli*, along with their substrates and the number of protein-protein interactions of each enzyme. “Char.” and “Unchar.” IPs are the numbers of characterized and uncharacterized interacting proteins, respectively. Compare to Figure 2.

Step	Enzyme	Uniprot	Full Name	Substrate	Char. IPs	Unchar. IPs
1	Pgi	P0A6T1	Glucose-6-phosphate isomerase	Glucose-6-phosphate (G6P)	8	0
2	PfkA	P0A796	ATP-dependent 6-phosphofructokinase 1	Fructose-6-phosphate (F6P)	19	0
2	PfkB	P06999	ATP-dependent 6-phosphofructokinase 2	Fructose-6-phosphate (F6P)	11	0
3	FbaA	P0AB71	Fructose-bisphosphate aldolase class 2	Fructose-1,6-bisphosphate (FBP)	9	0
3	FbaB	P0A991	Fructose-bisphosphate aldolase class 1	Fructose-1,6-bisphosphate (FBP)	4	0
4	TpiA	P0A858	Triosephosphate isomerase	Glyceraldehyde-3-phosphate (GAP)	22	0
5	GapA	P0A9B2	Glyceraldehyde-3-phosphate dehydrogenase A	Glyceraldehyde-3-phosphate (GAP)	52	4
6	Pgk	P0A799	Phosphoglycerate kinase	1,3-bisphosphoglycerate (1,3-BPG)	38	1
7	GpmA	P62707	2,3-bisphosphoglycerate-dependent phosphoglycerate mutase	3-phosphoglycerate (3-PG)	13	1
7	GpmI	P37689	2,3-bisphosphoglycerate-independent phosphoglycerate mutase	3-phosphoglycerate (3-PG)	1	2
7	GpmB	P0A7A2	Probable phosphoglycerate mutase	3-phosphoglycerate (3-PG)	1	0
8	Eno	P0A6P9	Enolase	2-phosphoglycerate (2-PG)	85	5
9	PykA	P21599	Pyruvate kinase II	Phosphoenolpyruvate (PEP)	20	2
9	PykF	P0AD61	Pyruvate kinase I	Phosphoenolpyruvate (PEP)	41	0

**Table 3 proteomes-09-00016-t003:** PPI data used in this study were obtained from these main 11 literature sources.

Scope	Source	PPIs (Count)	PPIs (%)	REF
*E. coli* complexes	Butland et al.	1	0.3	[25]
*E. coli* complexes	Hu et al.	148	42	[3]
Cell envelope complexes	Babu et al.	156	44.3	[26]
Binary PPIs (Y2H)	Rajagopala et al.	20	5.7	[2]
Bacterial Interactome comparison	Shatsky et al.	6	1.7	[27]
Bacterial inner membrane proteins	Bloois et al.	1	0.3	[28]
Chaperonin GroEL substrates	Houry et al.	2	0.6	[29]
Thioredoxin-targeted proteins	Kumar et al.	5	1.4	[30]
*Helicobacter pylori*/*E. coli* PPIs	Hauser et al.	8	2.3	[31]
*E. coli* complexes native/SDS-PAGE	Lasserre et al.	3	0.9	[32]
YajL and the thiol proteome	Le et al.	2	0.6	[33]

**Table 4 proteomes-09-00016-t004:** 12 overlapping PPIs coming from multiple large-scale sources.

Step	Enzyme	Uniprot ID	Interactor	Uniprot ID	Literature Source
2	PfkA	P0A796	MoeA	P12281	Butland/Rajagopala [25,2]
2	PfkA	P0A796	UcpA	P37440	Hu/Rajagopala [3,2]
3	FbaA	P0AB71	TreA	P13482	Hu/Shatsky [3,27]
5	GapA	P0A9B2	YidC	P25714	Babu/Bloois [26,28]
6	Pgk	P0A799	Usg	P08390	Rajagopala/Lasserre [2,32]
7	GpmA	P62707	NagC	P0AF20	Hu/Shatsky [3,27]
7	GpmA	P62707	GpmI	P37689	Rajagopala/Lasserre [2,32]
7	GpmI	P37689	GpmA	P62707	Rajagopala/Lasserre [2,32]
8	Eno	P0A6P9	Rne	P21513	Hu/Rajagopala/Shatsky [3,2,27]
8	Eno	P0A6P9	Pnp	P05055	Hu/Shatsky [3,27]
9	PykA	P21599	PflB	P09373	Hu/Rajagopala [3,2]
9	PykA	P21599	YggR (UP)	P52052	Hu/Rajagopala [3,2]

**Table 5 proteomes-09-00016-t005:** Summary statistics of this study.

Parameter	COUNT
Glycolytic enzymes	14
Unique protein-protein interactions (PPI)	339
Evidence based protein-protein interactions	58 (note A)
Unique protein interactors	237
Reproducible protein-protein interactions	13 (note B)
Uncharacterized proteins	14
PPIs involving uncharacterized proteins (UP)s	15 (note C)

Notes: (A) Evidence-based interactions have been reported in small-scale studies previously. (B) Reproducible PPIs found in 2 or more literature sources. (C) One of the uncharacterized proteins (YdjL; a probable zinc-type alcohol dehydrogenase) interacts with two glycolytic enzymes (GapA and Eno); hence the number of interactions involving UPs is 15.

**Table 6 proteomes-09-00016-t006:** Classification of interactors based on enzymatic or metabolic properties. NMPs are all other proteins not fitting into the first two choices. *Enzymes* are based on EC number, *Metabolic Protein* is based on GO and/or Metabolic Pathway (KEGG) annotation.

Classification	Enzyme	Metabolic Protein
Metabolic Enzyme (ME)	X	X
Metabolic Protein (MP)		X
Non-Metabolic Enzyme (NME)	X	
Non-Metabolic Protein (NMP)		

**Table 7 proteomes-09-00016-t007:** Potentially important uncharacterized interactors. These interactors are only known from high-throughput studies but their impact on glycolysis has not been validated independently. We selected a few examples based on abundance, conservation and their interaction in multiple (independent) studies (last column).

Glycolytic Enzyme	Uncharacterized Interactor	Uniprot ID	Filtering Criteria
PykA	YodC (60 aa peptide)	P64517	Top 10% abundance
Eno	YbcJ (70 aa peptide)	P0AAS7	Top 10% abundance
GapA	YiiD (an acyl transferase—may initiate fatty acid biosynthesis	P0ADQ2	Top 10% conservation
Pgk	YhhS (an MFS transporter-like YfeJ, an exporter of arabinose and herbicides	P37621	Top 10% conservation
Eno	YadG (part of an ABC exporter (TC# 3.A.1.105.17)ss	P36879	Top 10% conservation
GpmI	YegV (a sugar kinase)	P76419	Top 10% conservation
GapA	YdjL (a zinc ADH)	P77539	Multiple glycolytic interactor
Eno	YdjL (a zinc ADH)	P77539	Multiple glycolytic interactor
PykA	YggR a pilus biogenesis ATPase)	P52052	Found in two studies

## Data Availability

For data sources see the Materials and Methods section.

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
