# Peer review of "The Protein Interactome of Glycolysis in Escherichia coli"

_proteomes, 2021, doi:10.3390/proteomes9020016_

Round 1
Reviewer 1 Report
Review of "The protein interactome of glycolysis in Escherichia coli" by Chowdhury et al.
Dear authors,
In this work, the authors present a complete and comparative assessment of the glycolysis interactome in E. coli with recourse to evidence-based databases and try to identify unidentified and unlikely proteins which are likely to affect glycolysis. It is an interesting work that can be quite valuable if a few major and quite a few minor points are addressed. My requests for this review are below, divided into minor revisions and major revisions and are motivated by my own experience in biostatistics and conservation. I enjoyed that the authors put forward a few likely targets for this in the end of their work.
Minor revisions:
Introduction
- a lot of attention is dedicated in the introduction to the HPr, but I think this becomes exhausting after a while. While I understand the relevance, I do not think the reader needs such an extensive introduction on one specific agent
- in figure 1, there are very small labels below "Uncharacterized enzymes" and "Metabolic enzymes". If relevant, these should either be augmented or removed
- in lines 60-61, it is mentioned that there seem to be "few, if any, comprehensive analysis on the interface of ..."; I find this wording confusing, it sounds like the authors are not quite sure about this claim. If there are more studies they should be mentioned, if there are no more studies than this should be more explicitly stated (if mentioning uncertainty is important, "to the best of our knowledge, there are no studies..." would be more adequate)
- in lines 71-73 I would feel more comfortable if the authors cited more diverse literature (there are earlier papers doing experimental elucidation of allosteric of glycolysis mechanisms in E. coli)
Methods
-
- in line 124, "were" should be "was"
- there is a subsection missing (2.5)
Results
- in lines 171-172, the authors mention that "[t]hese numbers are roughly proportional to their total numbers of interactions" - could a quantification of this be provided?
- Figure 3 should have a more consistent colour scheme and, preferably, different colour schemes for different scopes of analysis (i.e. in Enzyme Interactors, one is faced with salmon for Enzyme on both, green and cyan for non-enzyme depending on the analysed plot, and blue for uncharacterised, but in Metabolic Interactors this colours refer to something which is completely different)
- In lines 207-209, the authors say "Note that the GO classification has a broad definition of “metabolism” so that processes like DNA repair are also considered “metabolic” (because they are involved in “DNA metabolism”)." Is such a broad definition relevant for this work?
- In the last parts of 3.1.3 the authors state that they find more metabolic than non-metabolic interactors were found, but then this is not verified by analysing KEGG - roughly, only half of the metabolic interactors is not involved in any metabolic pathways. Does this emerge from the author's earlier, definition of "metabolic" GO mentioned above or does it have other explanations pertaining to some incompleteness in the KEGG database?
- The lines in the violin plot are really quite wide, could they be made thinner?
- A subsubsection is missing (3.1.4)
- In line 293-294, the authors claim that "[e]ven if a protein binds to an enzyme with high affinity, the interactor can affect enzyme activity only when it is present at roughly stoichiometric (or higher) levels". Could the authors cite the relevant literature that would back this up? I know it is fairly intuitive but I would be more comfortable if this statement was backed up
- In Figure 6, is it not "Relative abundance" rather than "Abundance values"? Also, for better visualisation it would be good to jitter the points a little bit to the side in order to look properly at the data
Discussion
- In lines 325-327 it is stated that "replication and repair, translation and transcription as well as protein 325 folding, sorting and degradation are considered as (metabolic) pathways by KEGG, and 326 many of these processes could be regulating glycolysis or vice versa". How is this regulation of/by glycolysis? It would be helpful to better illustrate this by citing the relevant literature
- In Figure 7 the legends are misaligned, I ask the authors to please fix this
In general
- I think that nomenclature should be more uniform throughout this work - for example, "enolase", "Eno", "enolase (Eno)" and "eno" are used through this work. I understand that the three mentioned terms for enolase are interchangeable, but, for simplicity, the authors should have one symbol per concept or name and this should be consistent through the article - on its first instance in the text, it should be made clear what "enolase" will be referred to
- The authors should avoid stretching the images when building the Figures
Major revisions
Results
- I am not very comfortable seeing conservation quantified through a counting exercise when there are much better ways to do this... a more common way to quantify this would be through the calculation of some measure of entropy (Shannon's entropy, for example), rather than the presentation of a distribution. More particularly, one would be looking for measures such the Jaccard index (the ratio of the intersection and the union of two sets), which would allow the authors to calculate a metric for each interactor, and the relative entropy, which would allow the authors to compare the set of interactors for each enzyme with the "background" distribution (the distribution being used to calculate the average). The authors also claim that "glycolytic interactors seem to be better conserved than the average protein in the E. coli proteome" and I would want to see this better demonstrated (these criticisms extend to the discussion of conservation in the Discussion section and are, to some extent, addressed by addressing what has already been specified in this point). Additionally, a violin plot would be better suited to show continuous variables, whereas Figure 5 is a discrete count, so there may be better suited ways to show this
Discussion
- the authors state that they "predict that ~70% of glycolytic interactors are not present in adequate amounts to have a significant impact on glycolysis", but I see no way of making such a strong claim without the adequate modelling of the biochemical system. I understand this would be far too big of a request, so instead I would be more comfortable if the authors mentioned some relevant and more definite literature that would allow them to make this claim substantiated only by relative abundance. If no such literature exists, I think the statement should be revised to something that would focus more on what the authors state earlier, that "one of the goals of this study was to identify PPIs that are top targets for such quantitative measurements"
Reviewer 2 Report
The paper presents the study of proteins panel related to interactions (interactome) that influence glycolysis which is the one of the most important biochemistry catabolic processes.
The proteins involved in glycolytic interactions termed as interactors were studied in the E. Coli model organism and 339 PPTs.
The PPI dataset taken from the Uniprot database was used for the assessment of the E Coli glycolysis interactome.
REMARKS
ABSTRACT (note)
At the end, please refer more to the aim of this work. Stress more the outcome of studied glycolysis interactome – why it is important and what would be the impact. Do so, at the end of the Introduction.
INTRODUCTION (note, 1st paragraph)
I find quite important to start first paragraph involving some brief general description of proteomics and then related proteome study if interaction proteins called as interactome called as interactomics. Cite very recent works in this area. I do not see very representative to jump directily into the glycolysis process and therefore being “narrow” in the description.
INTRODUCTION (upgrade, 1st paragraph)
The systems biology subdiscipline which study interactions among proteins i.e. protein-protein interactions (PPTs), is called Interactomics [https://doi.org/10.1016/j.tibtech.2007.08.002]. However, the proteins must be analyzed via proteomic approach in the first place. Afterwards, a set of proteins referring to as protein panel, are collected and can be assigned to the molecular databases and pathway analyses [https://doi.org/10.1016/S0958-1669(00)00118-X]. Currently, the trend of the large-scale protein analysis heads to use autosampler systems to achieve very-high throughput performances [https://doi.org/10.1016/j.cca.2020.04.015]. After proteomics, the assessment of post-translational modifications and PPTs can be studied and assigned to interactome networks [https://doi.org/10.1016/j.tibtech.2007.08.002, https://doi.org/10.1016/j.bbapap.2020.140469].
Glycolysis is the primary central carbon metabolic pathway responsible for turning glucose …
MATERIALS AND METHODS, RESULTS, DISCUSSION (note)
This parts are exhaustive in length and fairly expressed. I value especially, the graphical part expressing particular glycolytic enzymes or interactomes.
Reviewer 3 Report
The manuscript of Chowdhury et al. regards an interesting phenomena, the hypothetical regulation of glycolytic flux by interaction of glycolytic enzymes with their protein (non-glycolytic) binding partners. The authors provide, based on a screen of several databases, a convincing evidence that only few the interactors may participate in the regulation of glycolysis activity and this is not unexpected since now it is a common believe that in most cases glycolytic enzymes interact with their partners to regulate other cellular phenomena, but not those related directly to just glycolysis.
One of the weakness of the manuscript is that the authors’ considerations are based on only one literature data on the quantitative proteomics (Schmidt, 2016) – as a matter of fact it was not the deepest uantitative study on E.coli proteome and the authors should justify their choice of the data source.
There are also some confusing points which should be corrected, e.g. Chapter 3.1. is entitled ‘Glycolytic enzymes interact very specifically’ but I could not find any information about specificity of the interactions in this chapter.
The authors should be also more consistent describing their findings, e.g. in Chapter 3.2. they wrote ‘Overall, glycolytic interactors seem to be better conserved than the average protein in the E. coli proteome’ whereas in Discussion (line 356) ‘Despite the almost universal conservation of glycolysis throughout all domains of life, many of the interactors considered in our analyses are much less well conserved’
I am also not sure what the authors wanted to show on Figure 6A.
And finally, do the interactions showed in Table 7 were those which had been confirmed experimentally? (as a physical interactions)
Round 2
Reviewer 1 Report
Dear authors,
I see that my concerns have been addressed, either through modifications to the manuscript or through helpful explanations and I would like to thank the authors for that. But I would strongly encourage the authors to not stretch Figure 6A. Apart from this minor change I think the manuscript sounds better now and is good for publishing.
Reviewer 2 Report
Authors have almost accomplished the remarks given.
However, a recent work about the state-of-the-art proteomics based on its automation should be added. Automation brings substantial improvement to obtain acceptable protein coverage within a short period of time, often required to evaluate disease in required time.
INTRODUCTION UPGRADE (1st paragraph)
Proteomics has produced invaluable data about the composition, abundance, and function of all proteins in cells and organisms that can be obtained in a high-throughput manner [1, https://doi.org/10.1016/j.cca.2020.04.015].
